# Mitogenomic Features and Evolution of the Nile River Dominant Tilapiine Species (Perciformes: Cichlidae)

**DOI:** 10.3390/biology12010040

**Published:** 2022-12-26

**Authors:** Yosur G. Fiteha, Mohamed A. Rashed, Ramadan A. Ali, Diaa Abd El-Moneim, Fahad A. Alshanbari, Mahmoud Magdy

**Affiliations:** 1Genetics Department, Faculty of Agriculture, Ain Shams University, Cairo 11241, Egypt; 2Department of Zoology, Faculty of Women for Art, Science and Education, Ain Shams University, Cairo 11566, Egypt; 3Department of Plant Production (Genetic Branch), Faculty of Environmental Agricultural Sciences, Arish University, El-Arish 45511, Egypt; 4Department of Veterinary Medicine, College of Agriculture and Veterinary Medicine, Qassim University, Buraydah 52266, Saudi Arabia

**Keywords:** Nile tilapiine, Haplotilapiine, mitochondrial genome, NGS, phylogeny

## Abstract

**Simple Summary:**

The Haplotilapiine lineage is a specific cichlid group that is widely distributed in the Nile River’s Egyptian sector, divided into 22 tribes with 3000 species, and constitutes one of the most diverse subclades. However, the systematics of Haplotilapiine and their evolutionary histories still suffer from knowledge gaps that should be resolved to increase their usefulness as a model system. Therefore, we sequenced, assembled, and characterized five complete mitochondrial genomes representing three Egyptian tilapiine fish species (*Coptodon zillii*, *Oreochromis niloticus*, and *Sarotherodon galilaeus*) dominating the Nile River waters. These mitogenomes have the typical circular mitogenome characteristics of 13 protein-coding genes, 22 transfer RNA genes, 2 ribosomal RNA genes, and 1 non-coding region. Our results indicate that the structure, nucleotide composition, synteny, and gene orders are the main conserved aspect of cichlids. The phylogenetic analyses using the maximum likelihood method indicate that species belonging to the Nile tilapiine appear paraphyletic and provide a further supplement to the scientific classification of fish. These results provide a comprehensive framework and worthy information for future research on this family.

**Abstract:**

To better understand the diversity and evolution of cichlids, we sequenced, assembled, and annotated the complete mitochondrial genomes of three Nile tilapiine species (*Coptodon zillii*, *Oreochromis niloticus*, and *Sarotherodon galilaeus*) dominating the Nile River waters. Our results showed that the general mitogenomic features were conserved among the Nile tilapiine species. The genome length ranged from 16,436 to 16,631 bp and a total of 37 genes were identified (two ribosomal RNA genes (rRNAs), 22 transfer RNA genes (tRNAs), 13 protein-coding genes (PCGs), and 1 control region). The ND6 was the only CDS that presented a negative AT skew and a positive GC skew. The most extended repeat sequences were in the D-loop followed by the pseudogenes (trnS^GCU^). The ND5 showed relatively high substitution rates whereas ATP8 had the lowest substitution rate. The codon usage bias displayed a greater quantity of NNA and NNC at the third position and anti-bias against NNG. The phylogenetic relationship based on the complete mitogenomes and CDS was able to differentiate the three species as previously reported. This study provides new insight into the evolutionary connections between various subfamilies within cichlids while providing new molecular data that can be applied to discriminate between Nile tilapiine species and their populations.

## 1. Introduction

African cichlid fishes (Perciformes: Cichlidae) are among of the most species-rich and phenotypically diverse families of vertebrates [1]. The African cichlid fish radiations are the most diverse extant animal radiations and provide a unique system to test predictions of speciation and adaptive radiation theory. The past few years have seen major advances in the phylogenetics, evolutionary biogeography, and ecology of cichlid fish [2]. According to the latest phylogeny, the Haplotilapiine lineage formerly referred to as “Tilapia” is a specific cichlid group that is widely distributed in the Egyptian sector of the Nile River, which is divided into 22 tribes with 3000 species, constituting one of the most diverse subclades [3,4,5,6]. Tilapiines are a widespread paraphyletic species applied to three genera of fish in the family Cichlidae: *Oreochromis*, *Coptodon*, and *Sarotherodon* [6,7,8,9]. They are endemic to Africa and considered the dominant freshwater species in Egypt, but interest in their aquacultural potential led to nearly worldwide distribution within the past fifty years [10]. Currently, behind carp, tilapia is listed as the second most farmed fish in the world yielding 4.5 MT in 2014, and it is expected to rise to 7.3 MT in 2020 [11]. The haplotilapiine species have become such an important global food source, not only because they are an inexpensive source of protein but also because of their advantages that are used in the aquaculture industry such as rapid growth, high environmental tolerance, fast spawning, relative resistance to poor water quality and disease, a high capacity to withstand a wide range of environmental stresses, and the presence of antinutrient compounds in their diet, making it an ideal target species for upgrading low-quality ingredients into high-quality food [12,13].

In recent decades, the mitochondrial genome (Mitogenome) of Metazoa has been considered the marker of choice for the reconstruction of phylogenetic relationships at several taxonomic levels, from populations to phyla, and has been widely used for the resolution of taxonomic controversies [14]. The mitochondrial genome is an extrachromosomal genome that has been found in most eukaryotes. Typically, an animal mitogenome is a double-stranded circular molecule, containing 13 protein-coding genes (PCGs), 22 transfer RNA genes (tRNAs), 2 ribosomal RNA genes (rRNAs), and 1 control region (CR), and the mitogenome length is approximately 15–18 kb [15]. Furthermore, the mitochondrial gene content is strongly conserved across a wide range of animal species with very few duplications, no intron, and very short intergenic regions [16]. The specific biological characteristics of the mitogenome such as the small size, simple structure, high mutation rate, the near absence of genetic recombination, uniparentally inherited, and evolving rapidly at the nucleotide sequence level have made it the most widely used marker for estimating genetic diversity among species. 

The phylogenetic diversity of Cichlid fish can be well-apprehended through the proper investigation of the mitochondrial evolutionary genomics [17]. The first complete mitochondrial genome for *Oreochromis niloticus* and *Oreochromis aureus* were determined in 2011, considering their economic importance in aquaculture [18]. As for the *Tilapia zillii* (*Coptodon zillii*) and *Sarotherodon galilaeus*, the complete mitochondrial genomes were announced in 2015 and 2021, respectively [19,20]. Several studies provide hypotheses for the phylogenetic placement of the cichlid radiations, however, none of these phylogenetic analyses were based on complete mitogenome sequences.

Egypt is one of Africa’s top aquaculture producers, accounting for two-thirds of the continent’s production and the third largest tilapia producer globally [21]. However, the systematics of Haplotilapiine and their evolutionary histories still suffer from knowledge gaps that should be resolved to increase their usefulness as a model system. Therefore, this study aimed to provide a comparative mitogenomic analysis at various taxonomic levels, and five mitogenomes representing three Nile tilapiine species (*Coptodon zillii*, *Oreochromis niloticus*, and *Sarotherodon galilaeus*) were sequenced, assembled, annotated, and combined with various Nile tilapiine species available from the GenBank database. Mainly, we focused on the molecular evolution of mitochondrial genes within genera and species. In addition, phylogenetic trees were reconstructed based on various mitochondrial gene sets to analyze the evolutionary relationships within the Cichlids family. The results will enrich the existing database, promote the study of the genetic recognition of the genus, and contribute to phylogenetic, evolutionary, developmental, conservation, and taxonomic investigations.

## 2. Materials and Methods

### 2.1. Sample Collection and DNA Extraction

Nile tilapiine species were collected, namely an *Oreochromis niloticus* specimen from the local fish market of Qarun lake (29.4840° N, 30.6545° E); a *Coptodon zillii* specimen from a local fish market near Nasser Lake (22.7395° N, 32.1973° E); and a *Sarotherodon galilaeus* specimen from a local fish market near Manzala Lake (31.3306° N, 32.0497° E), all from Egypt in June 2021. Species identification and assignment were independently performed using specific morphological descriptors [22]. Pieces of fish fins were preserved in absolute ethanol at room temperature until DNA extraction. Total genomic DNAs were extracted from fin tissue samples using GenElute™ Mammalian Genomic DNA Miniprep Kits (Sigma, Steinheim, Germany) with a final elution volume of 50 μL. Subsequently, the DNA extract was checked on 1% agarose gel, and DNA concentration and purity were determined with Quantus™ Fluorometer (Promega, Madison, WA, USA). Extracted DNA was stored at −20 °C until required.

### 2.2. Library Preparation Mitogenome Sequencing, Assembly, and Annotation

Shotgun sequencing libraries were prepared from each of the DNA extracts using the Illumina TruSeq library preparation kit following the manufacturer’s instructions (Illumina, San Diego, CA, USA). The libraries were sequenced using the Illumina HiSeq 4000 platform at Novogene (Beijing, China) with ~300 bp insert size at 11× sequencing dept. High-quality clean reads were filtered, and the de novo assembly was conducted using the single-contig approach [23]. The Nile tilapiine mitogenomes were annotated as circular molecules using the online tool Geseq [24] with default parameters. The tRNA-Scan SE web server [25] was used to verify tRNAs through their cloverleaf secondary structure and anticodon sequence. All the coding sequences were confirmed and corrected by translation using Geneious R10 [26]. A graphical map of the consensuses sequence of the mitogenomes was drawn using Geneious R10, combined with the CGView [27] to visualize the GC skews.

### 2.3. Sequence Analysis of the Nile Tilapiine Mitogenomes

In addition to newly sequenced samples, tilapiine mitogenomes from the GenBank database repository were included (Appendix A). All the mitogenomes were revised and filtered according to [4]. Each of the complete mitogenomes, protein-coding genes, and tRNAs, were separately aligned using the MAFFT algorithm [28]. To assess the genetic diversity of the Nile tilapiine species, the number of segregating sites (S), haplotypes (h), haplotype diversity (Hd), nucleotide diversity (π), and Tajima’s D [29] were estimated by DnaSP v.6 [30] using default settings. Strand asymmetry was assessed and calculated using the following formulas: AT skew = ((A − T) / (A + T)), and GC skew = ((G − C) / (G + C)) [31]. The codon usage frequency was estimated using MEGA 11 software [32] and relative synonymous codon usage values (RSCU) were determined which is an important measure of codon usage bias [33] and visualized using the RAWGraphs online tool [34].

Tandem repeats on the multiple sequence alignment of the Nile tilapiine mitogenomes were detected using Phobos V3.3 (http://www.rub.de/ecoevo/cm/cm_phobos.htm, accessed on 30 August 2022) with default settings for imperfect search mode in a unit size range from 2 bp to 15 bp, implemented in Geneious R10. To identify the single nucleotide polymorphisms (SNPs), the consensus sequence was set as the reference sequence, then the “Find Variations/SNPs” function was used per species through Geneious R10. The parameters were set as 1 for the minimum coverage, "an approximate" for the p-value calculation method and the reference sequence (only find variants in the sample) for the advanced parameters were ignored. Furthermore, the “compare annotations” function was applied as a default parameter to identify SNPs among species. 

An analysis of the evolutionary rates was performed for the genes shared by the Nile tilapiine species. In the beginning, the *Sarotherodon galilaeus* (NC_056897) was set as the reference genome. After that, the numbers of nonsynonymous (dN), synonymous substitutions (dS), and substitutions ratios (ω = dN/dS) were estimated for each mitogenome within the alignment of the coding nucleotide sequences of all Nile tilapiine species implemented in Geneious R10. Additionally, changes in amino acids were inferred at the protein level by classifying mutations according to their types in which the nonsynonymous substitutions were used and excluded the synonymous substitutions.

### 2.4. Phylogenetic Analysis 

To investigate the phylogenetic relationships between the newly sequenced species and additional Nile tilapiine representative species from the GenBank database (NCBI), phylogenetic analyses were performed based on the complete mitochondrial genome, CDS, D-loop region, and IGS using the maximum likelihood (ML) methods. The ML analysis was computed using FastTree V2 [35], performed under the generalized time reversible (GTR) model, using the default settings, implemented in Geneious R10. The published mitogenome of *Oncorhynchus keta* (GenBank: NC_017838) was set as the outgroup.

## 3. Results

### 3.1. Mitogenome Structure and Organization

A total of five complete mitogenomes of three Nile tilapiine species (*C. zillii, O. niloticus*, and *S. galilaeus*), were de novo assembled and successfully analyzed. The size of mitogenomes ranged from 16,436 bp (*O. niloticus*) to 16,631 bp (*S. galilaeus*; Appendix A). The structure of the mitogenomes was a typical circular DNA molecule, and the syntenies and gene orders exhibited the general mitogenome structure seen in cichlids (Figure 1). The Nile tilapiine mitogenomes consisted of 37 typical mitochondrial genes of which there were 13 protein-coding genes (11,474–11,473 bp), 22 transfer RNA genes (1554 bp), 2 ribosomal RNA genes (2638–2640 bp), and 1 control region (852–930 bp). Nine genes were encoded by the light strand (ND6 and eight tRNAs), and other genes were encoded by the heavy strand (28 genes: 14 tRNAs, 12 PCGs, and 2 rRNAs; Appendix A). The initiation codon used for all PCGS was ATG for methionine except GTG for CO1, and two stop codons were employed, namely TTA (87.5%) and TAG (12.5%). Incomplete stop codons were detected for CYTB, COX2, COX3, ND3, and ND4 (Appendix A). Two rRNA genes were found in the three species: the small subunit ribosomal RNA (12S) and the large subunit ribosomal RNA (16S), which were separated by trnV. The Nile tilapiine mitogenomes have the standard set of 22 tRNA genes, one for each amino acid and one additional isotype for each serine and leucine. All tRNAs were scattered around the circular mitogenome and varied in size from 66 bp (trnC) to 74 bp (trnL^UAA^ and trnK). All tRNAs had a similar folding pattern and secondary structure, with the exception of trnS^GCU^ which lacked the D-arm.

The analysis of the base composition of the Nile tilapiine mitogenomes showed that the average A, C, G, and T contents were 27.76%, 30.82%, 15.78%, and 25.66%, respectively. Overall, the three species’ base composition was very similar (Appendix A) and showed an obvious AT content preference which ranged from 52.80% (*C. zillii*) to 54.10% (*O. niloticus*), whereas the GC content ranged from 47.20% (*C. zillii*) to 46.60% (*S. galilaeus*; Appendix A).

To further estimate and understand the level of base bias between all genes, we calculated the AT and GC skew ratios for each gene in the mitogenomes of Nile tilapiine species (Figure 1). The results show that most of the AT skew of PCGs was variable: eight genes were negative, and two genes (ATP8 and COX2) were positive in all species. While the ND2 was symmetric in *C. zillii*, *S. galilaeus*, and positive in *O. niloticus*, ND4 was symmetric in *S. galilaeus*, *O. niloticus*, and positive in *C. zillii*, and ND5 was symmetric in *C. zillii*, *S. galilaeus*, and positive in *O. niloticus*. The AT skew of the 22 tRNAs, and two rRNAs were positive and negative in the control region. Moreover, in all species, the GC skews of the tRNAs, rRNAs, and the PCGs were negative, and the positive GC skew was only presented in the ND6 gene.

### 3.2. Polymorphism Assessment

#### 3.2.1. Tandem Repeats 

Tandemly repeated DNA sequences, with a unit length longer than 2 bp, were identified (Table 1). The repetitive sequences were unevenly distributed, mostly (82.89%) within intergenic regions (IGS) of the mitogenome, while the remaining 17.1% were located in the pseudogenes (trnS^GCU^). The most abundant class of repeats was trinucleotide (32.89%), followed by dinucleotide (25%), pentanucleotide (21%), tetranucleotide (17.11%), and hepta-nucleotide (3.95%) among the analyzed species. A total of nine tandem repeats were identified (Table 1), with lengths ranging from 8 bp to 14 bp, and the most extended repeat sequences were located in the D-loop. The pattern of repeat sequences within the Nile tilapiine mitogenomes was compared and found five repeated sequences across all three mitogenomes (TA/ CAA/ TTA/ CAA/ and AATAC).

Among these, the AATAC motif was found to be polymorphic between the three species, which was observed in *C. zillii*, and *S. galilaeus*, while the AATGC motif was observed in *O. niloticus*. Moreover, the TA motive was found to be polymorphic in the number of repeats (4–6) between different haplotypes of *C. zillii*. Four specific repeated sequences were explicitly identified for different species, two for *O. niloticus* (TG/ and TTATT), and two for *S. galilaeus* (TA/ and TTTAATT).

#### 3.2.2. Genetic Diversity

The results of nucleotide diversity (π) based on 37 mitochondrial genes from *C. zillii* mitogenomes show that the nucleotide diversity values ranged from 0.00 (eight tRNAs) to 0.1916 (ND6). For the *O. niloticus* mitogenomes, the nucleotide diversity values ranged from 0.00 (10 tRNAs) to 0.35 (ND4), while in *S. galilaeus*, the values ranged from 0.00 (17 tRNAs) to 0.014 (ND6). The haplotype numbers (Hap) and haplotype diversity (Hd) were estimated, and the results were presented in detail in Appendix A.

The number of segregating (S) sites for each gene of the Nile tilapiine mitogenomes was estimated (Appendix A). The highest number of S sites was recorded in *C. zillii* followed by *O. niloticus* and *S. galilaeus* in each of tRNAs and PCGs. The results demonstrate that the tRNAs were the most conserved genes among the three species, which ranged between 0 and 8, 3, and 1, respectively. In contrast, the ND complex contained the highest number of segregating sites which ranged from 9 to 203, from 35 to 126, and from 3 to 23, respectively. In rRNAs, the 16S rRNA was recorded with the highest number of segregating sites in the three species and the S site values was ranged between 8 (*S. galilaeus*) and 51 (*O. niloticus*). In tRNAs, the trnP and trnW were the only two genes that recorded a number of S sites among the three species, in which two sites were recorded in *C. zillii* and *O. niloticus*, while 1 in *S. galilaeus* in trnP. As for trnW, one S site was recorded in each of *C. zillii*, and *S. galilaeus*, and two S sites were recorded in *O. niloticus*. In the PCGs, the highest number of S sites in *C. zillii* was recorded in ND4 (203) followed by ND2 (145), in contrast to COX3 (2) followed by ATP8 (3). As for the *O. niloticus,* the highest number of S sites was recorded in ND5 (162) followed by ND4 (100) in contrast to ATP8 (2). The highest number of S sites in *S. galilaeus* was recorded in ND5 (23) followed by ND4 (17), in contrast to ATP8 (1). Remarkably, the number of S sites recorded in *S. galilaeus* was the lowest among all (Appendix A). 

#### 3.2.3. SNPs and Indels 

Single nucleotide polymorphism (SNP) sites and indels were detected between the Nile tilapiine mitogenomes across the respective coding DNA examined at the inter and intraspecific levels (Figure 2). Nor indels (insertions and deletions) were detected in CDS between the Nile tilapiine mitogenomes. Over 1437 SNPs were determined in these mitogenomes, and a set of filtered SNPs were identified as intra-specific SNPs which represent 84.30%, while 15.7% represent inter-specific SNPs. At the level of intra-specific, the highest number of SNPs was detected in *C. zillii* (575), followed by *O. niloticus* (565), and *S. galilaeus* (73). There were three types of inter-specific SNPs: (i) those represented between the three species but fixed in one species; (ii) others that showed genotyped within each species and shared between species; and (iii) those that were only represented between two species. Eleven SNPs were recorded in type (i), and five SNPs were recorded in type (ii). As for type (iii), 157 SNPs were identified between *C. zillii* and *O. niloticus*, 27 SNPs were identified between *O. niloticus* and *S. galilaeus*, and 24 SNPs were identified between *S. galilaeus* and *O. niloticus.* The distribution of these SNPs within and between species is shown in Figure 2. 

### 3.3. Molecular Evolution of CDS 

#### 3.3.1. Neutrality Test

Tajima’s D was used to detect the selection of intra- and inter-population for rRNAs, CDS, and tRNAs (Appendix A). The Tajima’ D test at the intra-specific level did not reveal any significant deviation from neutrality (*p* > 0.05), indicating an absence of selective forces operating on *C. zillii* and *S. galilaeus* populations. As for the *O. niloticus* population, Tajima’s D values ranged from −1.526 to 1.031 and were significant for ATP6, COX1, COX2, COX3, CYTB, and ND6. At the interspecific level, the Tajima D values ranged from −1.468 to 2.009, in which 36 genes showed no significant values and the exception was for trnI which deviated significantly from zero (2.009).

#### 3.3.2. Substitution Rates of Protein-Coding Genes

To better understand the nature of the evolutionary selection pressure in Nile tilapiine species, dN, dS, and dN/dS (ω) values were calculated for each protein-coding gene (Table 2). The dN/dS values of 12 out of 13 of PCGs were less than 1, demonstrating that these genes underwent the purifying selection in the process of evolution. The highest dN/dS ratio in *C. zillii* was found in ND6 (0.32) followed by ND3 (0.20) and ATP8 (0.18), as opposed to COX3 (0.01). In *O. niloticus*, the highest dN/dS ratio was for ATP8 (0.33), followed by ND6 (0.24), and ND3 (0.22), as opposed to COX2, COX3, and ND4L where their dN/dS values were equal to zero. For the *S. galilaeus*, the highest dN/dS ratio was found in ND6 (1), followed by CYTB (0.40), and ATP6 and ND2 which both had the same ratio of 0.33. Interestingly, the lowest dN/dS ratio in *S. galilaeus* was 0.00 which was found in ATP8, COX1, COX2, COX3, ND3, ND4, and ND4L. 

A comparative analysis of PCGs revealed that the dN of ND5 was the highest (average dN = 116.33), in contrast to ATP8 (average dN = 6.67). Additionally, ND5 was recorded as the highest value of dS (average = 18.33), in contrast to COX3 (average dS = 0.33). The substitution rate varied, with the dN/dS ratio average ranging from 0 to 0.52. The highest synonymous substitution rate (average dN/dS = 0.52) was observed for the ND6, whereas the lowest average of dN/dS (0.00) was noted for COX3 followed by ND4L (0.03).

#### 3.3.3. Amino Acid Change

Given the close relationship between the Nile tilapiine species, examining the amino acid change could shed light on how natural selection affects molecular evolution. The *Oncorhynchus keta* (NC_017838) mitogenome was set as the reference genome to quantify the changes in amino acids across the 13 protein-coding genes. After that, we excluded any variants that were only related to the reference genome, the variant frequency equal to 100%, and synonymous substitutions. A total of 177 nonsynonymous mutations were recorded between the Nile tilapiine species, and the highest number of amino acid changes was recorded in *C. zillii* (113), followed by *O. niloticus* (125), and *S. galilaeus* (104).

ND4 exhibited the highest number of amino acid changes among the Nile tilapiine species followed by ND5 and ND2, respectively. While the lowest number of changes was recorded in ATP8 followed by COX2 (Figure 3A). The number of changes in amino acids was similar in the three species in ATP8, CYTB, ND2, ND6, and ND4L. A total of 54 amino acid changes were detected: among which 8 changes were equal in the three different species, 7 changes were species-specific, 7 changes were identified in only two species, 8 changes were detected as forward mutation only, and 22 changes were detected as forward and reverse mutations (Appendix A). 

An imbalance between the forward and reverse mutation was observed in the amino acid changes of Valine (V) ↔ Isoleucine (I), Leucine (L) ↔ Methionine (M), Phenylalanine (F) ↔ L, Threonine (T) ↔ I, and Serine (S) ↔ T. Among the three species, the V ↔ I was balanced in *O. niloticus* and *S. galilaeus*, while in *C. zillii*, the I to V change rate was higher than V to I. The T ↔ I was balanced in *S. galilaeus* than *C. zillii* and *O. niloticus*, the latter two had higher T to I than I to T. The T ↔ S was balanced in *C. zillii*, and less balance in *O. niloticus*, followed by *S. galilaeus*. Additionally, the F ↔ L was unbalanced among the three species, the F to L recorded the highest rate of changes in *S. galilaeus* followed by *O. niloticus*, and *C. zillii*, while from L to F, the highest change rate was recorded in *O. niloticus*, followed by *C. zillii* and *S. galilaeus*. For L ↔M, the highest change from L to M was recorded in *S. galilaeus* followed by *O. niloticus*, and *C. zillii*, whereas from M to L the highest was recorded in *S. galilaeus*, and the same number of changes were recorded in *C. zillii* and *O. niloticus* (Figure 3B).

#### 3.3.4. Codon Usage Pattern

A total of 3823 codons were used for coding 20 amino acids by a standard set of 64 codons. The total number of available codons in the Nile tilapiine mitogenome varies from 3823 codons in *S. galilaeus* to 3824 in *C. zillii* and *O. niloticus.* The codon usage frequencies were generally similar among all mitogenomes. The amino acids Leucine and Serine were encoded by six different codons, while the rest of the amino acids were encoded by either two or four codons. Among amino acids, Leucine (17%) was utilized at the highest frequency followed by Alanine (9.1%), and Threonine (8.1%), in contrast to Cysteine (0.7%). The most frequently used codon was CUC (5.43%, Leu) followed by CUA (5.03 %, Leu) and AUU (4.48%, Ile), while GUG (0.026%, Met) was the least common codon.

The analysis of relative synonymous codon usage (RSCU) was performed in three stages (Figure 4). In the first stage, codons were clustered and ranked according to their RSCU values into three major groups namely: high (H), middle (M), and low (L); regardless of the codon and the amino acid (Figure 4A). The total average of RSCU values ranged from 0.15 to 1.93 across the 13 PCGs, in which codons in the subgroup H1 showed the highest values of RSCU, versus the L2 subgroup. The high group included H1 and H2 with an overall average of RSCU values ranging from 1.44 to 1.83 and from 1.12 to 148, respectively. Three subgroups (M1, M2, and M3) were included in the middle group with a total average of RSCU values that ranged from 0.97 to 1.15, from 99 to 1.10, and from 0.82 to 0.88, respectively. The Low group included L1 and L2 with an overall average ranging from 0.44 to 0.76, and 0.15 to 0.36, respectively. Noticeably, the codons ending in G have the lowest RSCU values that appeared in the L2 group (Figure 4A).

The second stage was based on the preferences of the codon to clarify the reason for the codon bias. Codons were arranged according to the position of the nucleotide in each of the three positions (first base, second base, and third base; Figure 4B). The highly biased frequencies were found for GCC (Ala), CUU (Leu), and CCU (Pro). An obvious bias was defined at the second base in ANA (Glu, Ala, Cys, and * ”Stop codon”), GNC (Arg, Pro, Met, and Gln), and CNC (Leu, His, Gly, and Lys). Both codons (GNC and CNC) had an exception in which the GUC (Arg) and CAC (Gly) codons showed no codon bias (RSCU value = 1) in the *O. niloticus*, and *C. zillii* species, respectively. While at the third base, the codon bias was defined in CUN^U/C/A^ (Leu), ACN^U/C/A^ (Ala), and AGN^C/A/G^ (Cys, Asp). A general bias existed against the G at the third codon, represented by these codons; UNG (Lys, Thr, Ser, Val), CNG (Leu, Ile, Gly, Leu), and GNG (Ser, Pro, Asn, Arg). Furthermore, codon bias was observed against G and U at the second and third bases (NGU). Two cases determined an exception in which all codons were biased against except for one group. Firstly, in ANG, the bias was recorded at the third base (AGG, Asp), and secondly, in GNU, the bias was recorded at the first base (GUU, Ser; Figure 4B). 

The unequal usage of synonymous codons in protein-coding genes was determined based on the frequencies of RSCU (Figure 4C). The analysis reveals that related species preserve the stability of codon usage behavior; as the use of one particular codon increases, the use of other synonymous codons decreases, implying a larger bias in occurrence. Variations in codon usage within and between the Nile tilapiine mitogenomes have been described to decipher the codon usage pattern. Across all mitogenomes, six synonymous codons were employed for Leucine (L) and Serine (S), and the most frequent codon for Leu was CUU (1.72) in contrast to UUG (0.72), while it was UCC (1.43) for Serine in contrast to UCG (0.34). Four synonymous codons were encoded for Alanine (A), Glycine (G), Proline (P), Arginine (R), Threonine (T), and Valine (V). The most abundantly used codons for these amino acids were GCC (1.86), GGC (1.54), CCU (1.7), CGC (1.49), ACA (1.39), and GUU (1.42); as opposed to GCG (0.11), GGU (0.67), CCG (0.19), CGG (0.62), ACG (0.23), and GUG (0.47), respectively. The rest of the amino acids were encoded by two codons. The most frequent codon for Cysteine (C) was UGC (1.42) in contrast to UGU (0.58), GAC (1.98) for Aspartic Acid (D) in contrast to GAU (0.80), GAA (1.08) for Glutamic Acid (E) in contrast to GAG (0.14), UUU (1.06) for Phenylalanine (F) in contrast to UUC (0.94), CAC (1.73) for Histidine (H) in contrast to CAU (0.26), AUU (1.28) for Isoleucine (I) in contrast to AUC (0.72), AAA (1.51) for Lysine (K) in contrast to AAG (0.17), AUA (1.15) for Methionine (M) in contrast to AUG (0.85), AAC for Asparagine (N) in contrast to AAU (1.07), CAA (1.69) for Glutamine (Q) in contrast to CAG (0.31), UGA (1.36) for Tryptophan (W) in contrast to UGG (0.64) and UAU (1.45) for Tyrosine (Y) in contrast to UAC (0.99). 

Comparative analysis showed that most codon usage patterns and the major customarily utilized codons of the three species were conservative, but five variations were identified between the three species which lie in the RSCU value of codon bias (Figure 4C). The UUC (F) codon was identified as the most frequent codon in both *C. zillii* and *S. galilaeus* instead of UUU in *O. niloticus*. Codons such as CUC (L) and AAC (N) were the most abundantly used codons in *S. galilaeus* instead of CUU and AAU, respectively. While UCU (S) was the most frequent codon in *C. zillii* instead of UCC and ACC (T) in *O. niloticus* instead of ACA. In the other two species. In addition, two variations were detected for the total RSCU value of all the Nile tilapiine mitogenomes, but the codon bias differed within the species. The most abundantly used codon in all mitogenomes was UAC (Y) instead of UAU, and CAU (H) instead of CAC in each of *C. zillii*, *O. niloticus*, and *S. galilaeus*. 

### 3.4. Phylogenetic Analysis

In the present study, the phylogenetic relationships were analyzed based on the sequences of the complete mitogenome, the CDS, the D-loop region, and IGS to clarify the relationships in Nile tilapiine species (Figure 5). The newly sequenced species and other previously published Nile tilapiine species were analyzed using the maximum likelihood method and rooted by *Oncorhynchus keta*. In all four trees constructed, species belonging to the Nile tilapiine appear paraphyletic and all major clades within the trees were well supported with high bootstrap values. 

The phylogenetic status based on the complete mitogenomes and CDS were highly similar to the published taxonomic information (Figure 5A,B). Whereas the phylogenetic topologies clearly showed that each species was grouped into distinct clear clades and the *O. niloticus* was phylogenetically closer to *S. galilaeus*. The phylogenetic status based on the D-loop region and IGS were not similar to the published taxonomical information (Figure 5C,D). In detail, the tree was constructed based on the D-loop region showed that the *S. galilaeus* samples were correctly clustered with the matched species at high bootstrap support (=1). However, both D-loop and IGS phylogenetic trees were unable to group samples that represent *O. niloticus* and *C. zillii* with other members of the species in the same clade.

## 4. Discussion 

Here, we report the first comparative analysis based on mitogenome-wide variability patterns in the dominant Nile tilapiine species belonging to the Haplotilapiine lineage (*C. zillii*, *O. niloticus*, and *S. galilaeus*), which will be an essential addition to cichlids genomic resources and can help distinguish between forces affecting the evolution of these mitogenomes. In line with expectations, the comparative genomic analysis between the Nile tilapiine and cichlids revealed a highly conserved synteny [19,20,36]. 

The overall base composition was very similar for the three species, and to what has been reported for haplotilapiine species [18,19,36]. The base composition results displayed an excess of C over G and A over T. The lower value for GC compared to AT is another common feature that has been observed in most vertebrate mitogenomes resulting from anti-bias against G in the third codon position [37]. The AT-skew and GC-skew were considered potential indicators to measure the strand asymmetry in which the nucleotide skew might be attributed to the equilibrium between mutation and selection pressure during replication and transcription [38,39,40]. The ND6 showed the highest fluctuation, which recorded the lowest AT-skew and highest GC-skew values, suggesting that the selection and mutational pressure on this gene might be significantly different from other genes. Notably, ND6 had a deviation range highly consistent with other studies [41,42].

Dispersed repeats are essential in generating genetic diversity and valuable contributions to evolution [43]. Here, the most extended repeat sequences were located in the D-loop followed by the pseudogenes (trnS^GCU^), a common result in different organisms [44]. Four tandem repeats were identified at the intra-specific level, and according to their variability, these repeats could be used as markers in population genetic analyses [45]. Studies differed in explaining the origin of tandem repeats, one reason might be that mitochondrial DNA replication seems to be the primary mechanism causing the gain or loss of repeats through slippage-strand mispairing. The other was natural selection in evolutionary history, but it was difficult to distinguish one from the other [46,47].

As was expected, no indels were identified in CDS, where indels generally are under stronger purifying selection than SNPs; this is likely the result of the strongly deleterious effect of frameshift indels as the selection strengths on nonsynonymous SNPs and in-frame indels are not significantly different [48]. Although a few SNPs were identified at the intra-specific level, these variations are normal since different populations can experience different selective pressures according to the environment. Furthermore, these substitutions may have occurred due to random events in these populations [49]. This included some fixed SNPs between only two species and 11 SNPs that were common to all species and fixed in one species, which provided a potential genetic marker that can be used for population genetics studies. Additionally, several SNPs were uniquely recorded for each mitogenome, as these loci reflect the species’ history and population dynamics which could be promising for molecular systematics [23]. Noticeably, the ATP8 had fewer polymorphisms than the gene COX1 which is known for its great potential as a molecular marker in evolutionary studies [50].

Tajima’s D test is a widely used neutrality test in population genetics solely based on the distribution of allele frequencies or the site frequency spectrum [51]. Tajima’s D neutrality test for the *C. zillii* and *S. galilaeus* based on CDS regions showed negative insignificant values which indicated an excess of low-frequency polymorphism. As for *O. niloticus*, four genes were found to significantly deviate from neutrality (ATP6, COX1, COX2, COX3, CYTB, and ND6). At the inter-specific, Tajima’s D slightly fluctuated across rRNAs, CDS, and tRNAs but did not significantly deviate from zero, indicating no deviation from the neutral model of evolution, whereas the positive Tajima’s D was only detected in trnI in accordance with a previous report [4]. Positive values indicated a balancing selection but can also result from demographic processes, such as bottlenecks and population structure [52]. In fact, Tajima’s D has less power in detecting both positive and negative selection because of the transient nature of positive selection and the weak signal left by negative selection [53]. Therefore, the dN/dS ratio was used because it can detect a selection even if the overall level of variation is low. Our findings based on the average ratio indicated that all protein-coding genes evolved under purifying selection. This indicates that various genes evolved under natural selection. Purifying selection has been shown to be the main driver shaping mitogenome diversity [54,55,56]. However, ND6 exhibited the highest evolutionary rate and COX3 as the most conserved gene. Similar results were obtained in several studies of the mitogenomes analysis of African cichlids [36,57,58]. Thus, we can conclude that the ND6 gene may have evolved more rapidly than any other protein-coding genes among these African cichlids.

Evolution drives changes in a protein’s sequence over time [59]. The extent to which these changes in sequence lead to shifts in the amino acid was an essential question about it affecting the protein function. The greatest amino acid changes were found in the ND4 and ND5 genes, which were generally considered to be among the fastest evolving mitogenome genes in comparisons between closely related species. ATP8 recorded the lowest number of changes among the three species, suggesting that the shortest protein-encoding mitochondrial gene might not be informative at these or lower levels of divergence. The greatest change in amino acids was recorded towards Valine from Isoleucine followed by Isoleucine from Valine, and these changes were recorded in the three species. A previous study highlighted that Isoleucine has an important evolutionary aspect: sharing the codon’s first and second nucleotide with methionine, an important initiation codon for open reading frames [23].

Codon usage bias is an important evolutionary feature in a genome and has been widely documented in many organisms from prokaryotes to eukaryotes [60,61]. Thus, the comparative codon usage analysis facilitates understanding the evolution and adaptation of living organisms. In our analysis, the synonymous codons for similar amino acids did not appear at an equal frequency [62] and the alternative synonymous codons were similar among closely related species. The most frequently employed amino acid was Leucine followed by Alanine in contrast to Cysteine, an observation consistent with the previous work on *Cirrhinus reba* [63]. RSCU is commonly used to reflect codon bias, which removes the effect of the amino acid composition of a codon [64]. The highly biased frequencies were found for GCC (Ala) and CUU (Leu) [65], whilst the preferred codons usually occur at higher frequencies in the sequence of highly expressed genes than weakly expressed genes. Several studies have shown that various biological factors are involved in synonymous codon usage bias, such as the gene expression level [66], gene length [67], tRNA abundance [68], mutation bias and GC composition [69]. In the present study, 27 RSCU values were identified with values greater than 1, 11 of which were revealed as “over-represented”, which implied engaging with highly expressed genes for efficient protein synthesis via translational selection. The similar RSCU values for each amino acid in the three species suggested that the gene function in the Nile tilapiine is expressed similarly. They displayed a greater quantity of NNA and NNC, echoing what we previously mentioned about the nucleotide composition of the third position, for which there was a preference for A, C, and anti-bias against G, in Nile tilapiine as well as other fish species [60,61]. Directional mutation pressure and natural selection forces are major factors affecting codon bias in various organisms and have been widely used for inter-specific and intra-genomic codon usage variation [70].

The phylogenetic analysis classified the three species into three groups, based on whole mitogenomes and CDS regions. Our results were consistent with the basis for African cichlid phylogenetics and systematics [6]. Nevertheless, the D-Loop being more variable than the rest of the mitochondrial genome is, therefore, a very useful marker for the study of very divergent populations or species [71]. 

## 5. Conclusions

This work produced the complete mitogenome sequences of three species (*Coptodon zillii*, *Oreochromis niloticus*, and *Sarotherodon galilaeus*) which represent the most abundant and dominant cichlids in the Nile River in Egypt. Through comparative sequence analysis, the structure of the Nile tilapiine mitogenomes was a typical circular structure, and synteny reported in cichlids. The highest divergence in all parameters had been recorded in the *C. zillii*, which diverged from both *O. niloticus* and *S. galilaeus*. The molecular parameters and genetic variability divergence followed that pattern; most of the mutations were in ND genes and the fewest were in ATP8 followed by COX genes. This study reveals part of the radiant diversity of the cichlid freshwater fish family and provides valuable information to understand the coordinated evolution of the mitogenomes of Nile tilapiine and provides a core for the analyzing the cichlids.

## Figures and Tables

**Figure 1 biology-12-00040-f001:**
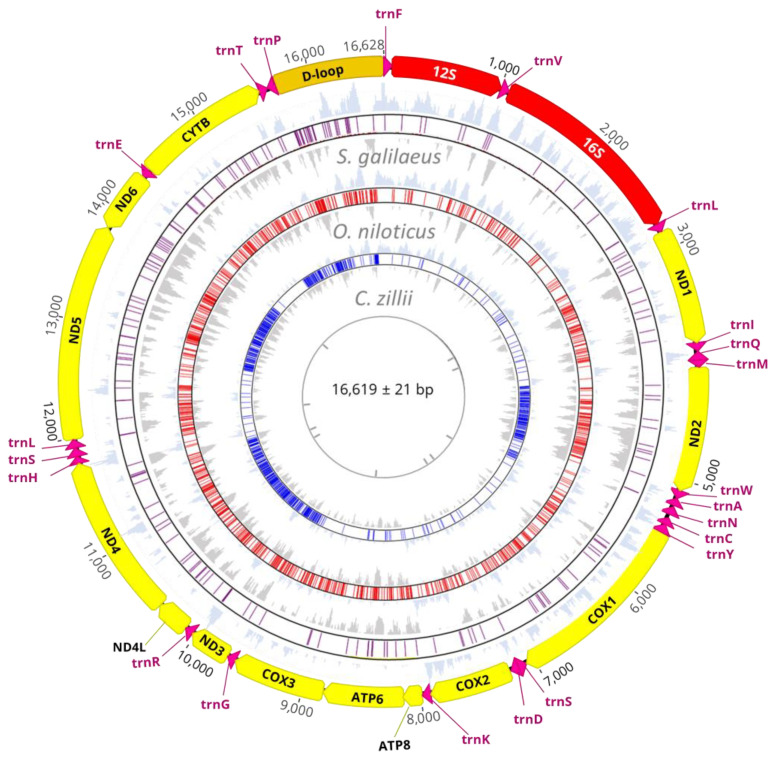
Genetic features and organization of the complete mitochondrial DNA sequence of the Nile tilapiine mitogenomes. The label’s color represents mitogenomes genes: CDS (yellow), tRNAs (dark pink), rRNAs (red), and the control region (D-loop; gold). The direction of genes is indicated by the arrows, and the genes with the clockwise arrows are encoded on the heavy strand, while the genes with the anticlockwise arrows are encoded on the light strand. SNP distribution in all genes is described in each species with highlighted tracks with different colors and laminated by tracks representing GC skews. Purple color represents C. zillii, red represents O. niloticus, blue represents S. galilaeus, gray represents SNPs shared between species, light gray represents GC skews (+), and light blue represents GC skews (−).

**Figure 2 biology-12-00040-f002:**
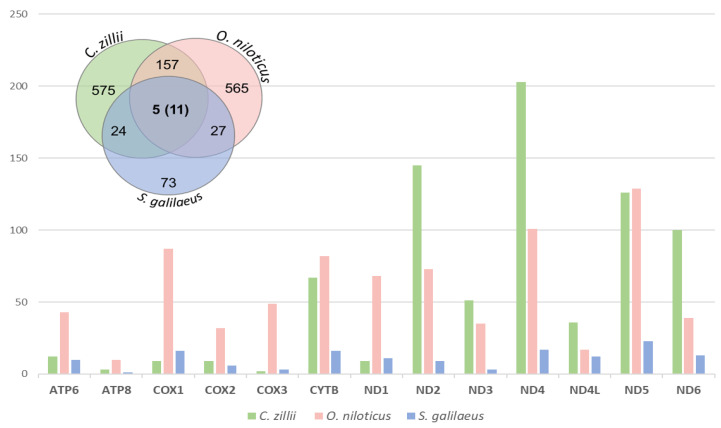
A comparative Vann diagram between three species of Nile tilapiine based on the number of SNPs annotated in CDS. The numbers exclusive to each circle show the number of SNPs that are polymorphic within one species and the number in the overlap area is that of the polymorphic sites shared between species. The distribution of the number of SNPs per CDS gene among the Nile tilapiine mitogenomes that represents in the clustered column chart. The light green color represents *C. zillii* species, the rose color represents *O. niloticus*, and the light blue color represents *S. galilaeus*.

**Figure 3 biology-12-00040-f003:**
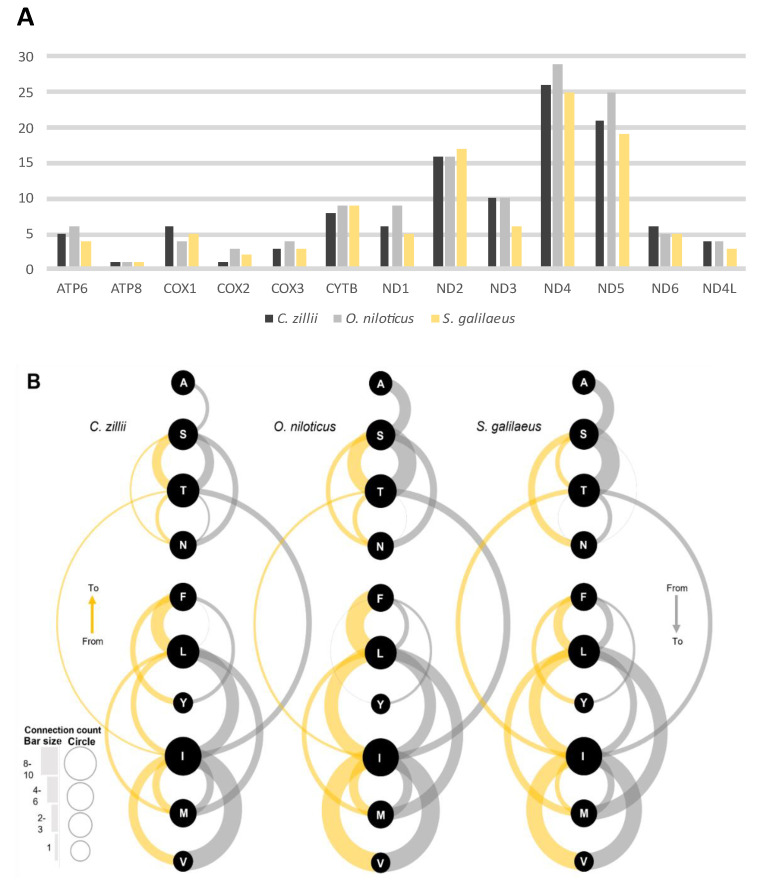
(**A**) Amino acid changes between the Nile tilapiine mitogenomes for the protein-coding genes. (**B**) The arc diagram displays the changes in amino acids among the Nile tilapiine species. Each change represents by a black circle and connects with arcs. The gray arcs visualize forward mutations, and the gold arcs visualize reverse mutations. Arc thickness and circle size represent the number of changes for these amino acids. The amino acids are described with a one-letter abbreviation.

**Figure 4 biology-12-00040-f004:**
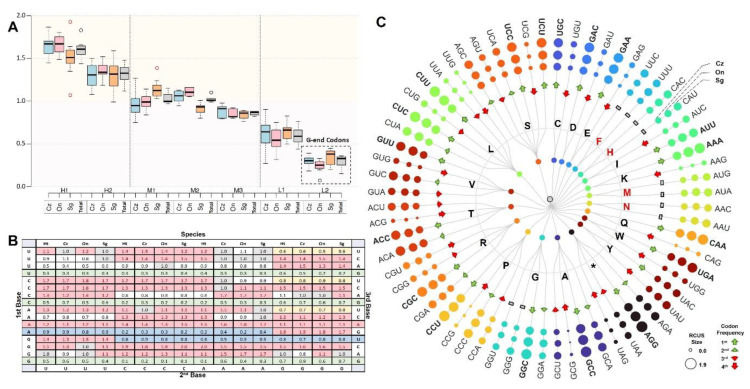
(**A**) The box plot representing the clustering analysis of codon usage bias based on the RSCU values. The highest RSCU values clustered in H1, while the lowest in L2. Each color represents a species, blue represents *C. zillii*, red represents *O. niloticus*, orange represents *S. galilaeus*, and gray represents the total average of RSCU values for the three species. (**B**) The relative synonymous codon usage (RSCU) patterns of 13 PCGS between the Nile tilapiine species. Each color represents several codons based on the RSCU values: red represents codon bias RSCU values > 1, gray represents codons showing no bias RSCU values =1, green represents codons’ relative rarity RSCU values <1, yellow represents codons bias against specific codons, and light blue represents codons bias against all codons except one codon. (**C**) The circular dendrogram represents synonymous codon usage for coding 20 amino acids. The amino acids are described with a one-letter abbreviation. The size of the circles represents the RSCU value that ranged from 0.00 to 1.90. Biased codons are marked with the green arrow (more abundant), while biased-against codons are marked by the red arrow (less abundant), and codons that showed no bias are marked by a gray rectangle.

**Figure 5 biology-12-00040-f005:**
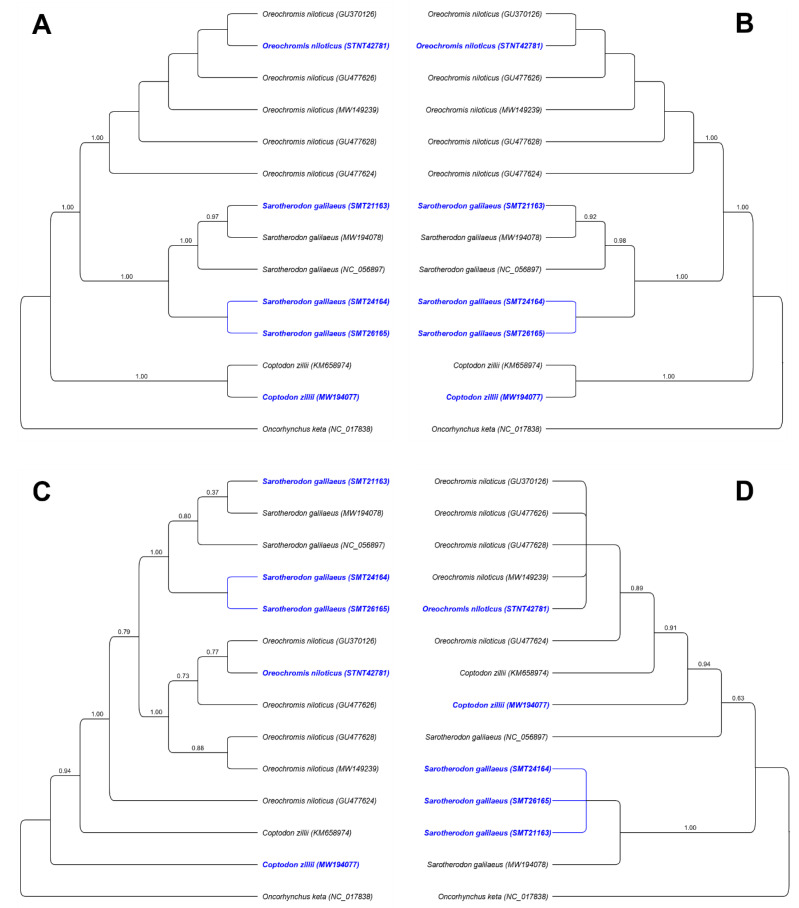
Maximum likelihood inference-based phylogeny showing the mitogenomic relationship using (**A**) complete mitogenome, (**B**) CDS, (**C**) D-loop, and (**D**) IGS, between the Nile tilapiine species (*C. zillii*, *O. niloticus*, and *S. galilaeus*). Samples under study are highlighted with a different color. Each label includes the species name and the GenBank accession number. The numbers at the nodes indicate the bootstrap values.

**Table 1 biology-12-00040-t001:** Tandem repeats of the Nile tilapiine mitogenomes, including repeat class, repeat unit, repeat abundances, percentage abundance, and location. The numbers in parentheses in the repeated unit represent this unit in each species *C. zillii*, *O. niloticus*, and *S. galilaeus*, respectively.

Repeat Class	Repeat unit (Cz, On, Sg)	Repeat Abundances	Percentage Abundance (%)	Location
Dinucleotide	TA (2, 6, 10)	18	23.68	D-loop
	TG (0, 1, 0)	1	1.32	D-loop
Trinucleotide	CAA (2, 6, 5)	13	17.11	trnS2 ^1^
	TTA (2, 5, 5)	12	15.79	D-loop
Tetranucleotide	CCCG (2, 6, 5)	13	17.11	trnN-trnC
Pentanucleotide	AATGC (0, 5, 0)	5	6.58	D-loop
	AATAC (1, 0, 5)	6	7.89	D-loop
	TTATT (0, 5, 0)	5	6.58	D-loop
7-nucleotide	TTTAATT (0, 0, 3)	3	3.95	D-loop
Total	-	76	100	-

^1^: Pseudogene.

**Table 2 biology-12-00040-t002:** The substitution rate of 13 protein-coding genes among the Nile tilapiine mitogenomes, the number of nonsynonymous substitutions (dS), the number of synonymous substitutions (dN), and the ratio (dN/dS).

CDS	*C. zillii*			*O. niloticus*			*S. galilaeus*		
	dS	dN	ω	dS	dN	ω	dS	dN	ω
ATP6	90	4	0.04	42	6	0.14	3	1	0.33
ATP8	11	2	0.18	9	3	0.33	0	0	0
COX1	159	7	0.04	105	7	0.07	2	0	0
COX2	66	7	0.11	41	0	0.00	0	0	0
COX3	73	1	0.01	46	0	0.00	1	0	0
CYTB	161	7	0.04	60	9	0.15	5	2	0.40
ND1	113	10	0.09	69	7	0.10	5	1	0.20
ND2	159	21	0.13	41	4	0.10	3	1	0.33
ND3	50	10	0.20	18	4	0.22	1	0	0
ND4	240	19	0.08	50	9	0.18	7	0	0
ND4L	45	4	0.09	11	0	0.00	1	0	0
ND5	224	30	0.13	118	24	0.20	7	1	0.14
ND6	76	24	0.32	25	6	0.24	3	3	1.00

## Data Availability

The DNAseq raw data used for the assembly of the reported mitogenomes as well as the full mitogenome sequence were deposited into the NCBI repository under Bioproject no. PRJNA684015 and SRA accessions no. SRR15131195 (*C. zillii*), SRR22809076 (*O. niloticus*), and SRR22808829-31 (*S. galilaeus*).

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
