# Peer review of "Mitogenomic Features and Evolution of the Nile River Dominant Tilapiine Species (Perciformes: Cichlidae)"

_biology, 2022, doi:10.3390/biology12010040_

Round 1

Reviewer 1 Report

In the manuscript of Fiteha et al. the authors present several newly sequenced mitochondrial genomes of three tilapiine species. They studied thoroughly the  genomic variation  with the standard tools and analyses typical for this kind of studies to better distinguish between three most abundant genera of ciclids in the Nile river.

My main claim, especially for the introduction, is the need to put this particular study into the historical context: if the mtDNA of these species was ever studied before; for which tilapian species the mitochondrial genomes were already sequenced and studied, and which species are lacking, etc. 

It is quite confusing that the authors claim that their comparative analysis of tilapiine genomes is the first (for example in line 433), while similar studies already exist (for example in the bibliography of the manuscript - 17, 57, 58).

In lines 174-175 I do not see much sense in average values of AT vs. GC content among three genera. Why not do it among all available mitogenomes of ciclids/tilapiine? Or just report the range within each of three species. Average among three species is quite arbitrary and not informative.

In section 2.2.2 it is not clear which kind of measure for genetic diversity did you use, if it is pi or proportion of heterozygotes or anything else.

In line 222 and in overall text the Latin names of the species should be written in italic. 

Line 226 - it is more infromative to give the ranges of the number of segregating sites in each species than averaged among these arbitrary three genera.

Line 247 - it is not clear what did you mean under the point ii and what is the difference between SNPs in the point i and ii.

line 326 -  (To → From) does not make sense and does not correspond to the figure. It is better just to write: The gray arcs visualize forward 326 mutations, and the gold arcs visualize reverse mutations.

Line 477 - better use "significantly deviated from neutrality" instead of "unneutral". In the same line the expression "coding genes" does not make sense, as there is no such thing as "non-coding genes".

Line 480 - "positive selection" or "positive Tajima's D"?

Line 487 - "functional genes" does not make sense, as all genes are functional by definition.

Line 537 - incorrect use of the word "popular". Should be substituted by "abundant", if it is the case.

Author Response

Reviewer 1

Dear Reviewer,

Thank you very much for your effort and time handling our manuscript. Please find our response to the reviewer’s comment as follows:

In the manuscript of Fiteha et al. the authors present several newly sequenced mitochondrial genomes of three tilapiine species. They studied thoroughly the  genomic variation  with the standard tools and analyses typical for this kind of studies to better distinguish between three most abundant genera of ciclids in the Nile river.

My main claim, especially for the introduction, is the need to put this particular study into the historical context: if the mtDNA of these species was ever studied before; for which tilapian species the mitochondrial genomes were already sequenced and studied, and which species are lacking, etc. 

Response: Thank you for the comment, we have added additional text as recommended.

It is quite confusing that the authors claim that their comparative analysis of tilapiine genomes is the first (for example in line 433), while similar studies already exist (for example in the bibliography of the manuscript - 17, 57, 58).

Response: Thank you for the comment, what we meant when we said that it is the first comparative analysis based on the complete mitochondrial genome is that it was the first time applying this comparative analysis for these species (Coptodon zillii, Oreochromis niloticus, and Sarotherodon galilaeus) in the same study, and samples were from the same country, and that's because these species are considered to be dominant in the Nile river in Egypt.  Previously, each species was studied separately, and the study for both C. zillii and S. galilaeus was a short article.

In lines 174-175 I do not see much sense in average values of AT vs. GC content among three genera. Why not do it among all available mitogenomes of ciclids/tilapiine? Or just report the range within each of three species. Average among three species is quite arbitrary and not informative.

Response: We appreciate the suggestion very much, noted and changed it accordingly.

In section 2.2.2 it is not clear which kind of measure for genetic diversity did you use, if it is pi or proportion of heterozygotes or anything else.

Response: Thank you for the comment, details about the genetic diversity estimated in either 2.2.2 (changed to 3.2.2) or other points were given in lines 122-124 in the M&M section. The 1st paragraph showed the pi diversity (nucleotide diversity) which we added the sign in the revised version followed by the segregating sites (s).

In line 222 and in overall text the Latin names of the species should be written in italic. 

Response: Thank you for the comment, noted and changed it accordingly.

Line 226 - it is more infromative to give the ranges of the number of segregating sites in each species than averaged among these arbitrary three genera.

Response: Thank you for the comment, noted and changed it accordingly.

Line 247 - it is not clear what did you mean under the point ii and what is the difference between SNPs in the point i and ii.

Response: Thank you for the comment, in point i, the SNP site was polymorphic between the three species but fixed in one species, while in point ii, the SNP site was polymorphic inside species and between species.

line 326 -  (To → From) does not make sense and does not correspond to the figure. It is better just to write: The gray arcs visualize forward 326 mutations, and the gold arcs visualize reverse mutations.

Response: Thank you for the comment; we noted it and changed it accordingly.

Line 477 - better use "significantly deviated from neutrality" instead of "unneutral". In the same line the expression "coding genes" does not make sense, as there is no such thing as "non-coding genes".

Response: We appreciate the suggestion very much, noted and changed it accordingly.

Line 480 - "positive selection" or "positive Tajima's D"?

Response: Thank you very much for the comment, we are sorry to use the mistake word.

Line 487 - "functional genes" does not make sense, as all genes are functional by definition.

Response: Thank you for the comment, we noted it and changed it accordingly.

Line 537 - incorrect use of the word "popular". Should be substituted by "abundant", if it is the case.

Response: Thank you for the comment, noted and changed it accordingly.

Reviewer 2 Report

This study produced the complete mitogenome sequences of three species (Coptodon zillii, Oreochromis niloticus, and Sarotherodon galilaeus) which represent the most popular cichlids in the Nile River in Egypt. This study reveals part of the radiant diversity of the cichlid freshwater fish family and provides valuable information to understand the coordinated evolution of the mitogenomes of Nile tilapiine and provide a core for the analysis for the cichlids. However, this article needs to add the following analysis

1The annotation of the mitochondrial genome of these three species should be presented in tables for clearer expression.

2The tRNA structures in these mitogenomes should be predicted in figures.

Author Response

Reviewer 2

Dear Reviewer,

Thank you very much for your effort and time handling our manuscript. Please find our response to the reviewer’s comment as follows:

This study produced the complete mitogenome sequences of three species (Coptodon zillii, Oreochromis niloticus, and Sarotherodon galilaeus) which represent the most popular cichlids in the Nile River in Egypt. This study reveals part of the radiant diversity of the cichlid freshwater fish family and provides valuable information to understand the coordinated evolution of the mitogenomes of Nile tilapiine and provide a core for the analysis for the cichlids. However, this article needs to add the following analysis:

1、The annotation of the mitochondrial genome of these three species should be presented in tables for clearer expression.

Response: Thank you for the comment, please kindly note that annotations were reported in supplementary table 2, also we mention that in lines 167 and 170 (revised version).

2、The tRNA structures in these mitogenomes should be predicted in figures.

Response: We appreciate the suggestion very much. We have previously studied the tRNA genes and their evolutionary dynamics in cichlids separately and then compared the whole family to the specific species (Coptodon zillii, Oreochromis niloticus, and Sarotherodon galilaeus) that belonging to the haplotilapiine lineage. We will be pleased to see our article entitled "The Evolutionary Dynamics of the Mitochondrial tRNA in the Cichlid Fish Family".

Fiteha, Y.G.; Magdy, M. The Evolutionary Dynamics of the Mitochondrial TRNA in the Cichlid Fish Family. Biology 2022, 11, 1522, doi:10.3390/biology11101522.
